# Degradation of Proteoglycans and Collagen in Equine Meniscal Tissues

**DOI:** 10.3390/ijms25126439

**Published:** 2024-06-11

**Authors:** Julia Dubuc, Melodie Jil Schneider, Valerie Dubuc, Helene Richard, Maxime Pinsard, Stephane Bancelin, Francois Legare, Christiane Girard, Sheila Laverty

**Affiliations:** 1Comparative Orthopedic Research Laboratory, Department of Clinical Sciences, Faculté de Médecine Vétérinaire, Université de Montréal, 3200 Sicotte Saint-Hyacinthe, Quebec, QC J2S2M2, Canada; juliadubuc@gmail.com (J.D.); melodie.schneider@sund.ku.dk (M.J.S.); valerie.dubuc@umontreal.ca (V.D.); helene.richard@umontreal.ca (H.R.); girardvaillancourt@gmail.com (C.G.); 2Institut National de la Recherche Scientifique, Université du Quebec, 1650 Bd Lionel-Boulet, Varennes, Quebec, QC J3X1P7, Canada

**Keywords:** meniscus, collagen, proteoglycan, intrasubstance, degradation, osteoarthritis

## Abstract

Investigate meniscal extracellular matrix degradation. Equine menisci (*n* = 34 from 17 horses) were studied. Site-matched sections were cut and scored from three regions (ROIs; *n* = 102) and stained for histology, proteoglycan (safranin O and fast green), aggrecan, and collagen cleavage (NITEGE, DIPEN, and C1,2C antibodies, respectively). Picrosirius red and second harmonic generation microscopy were performed to investigate collagen ultrastructure. A total of 42 ROIs met the inclusion criteria and were included in the final analysis. The median (range) ROI histological score was 3 (0–9), providing a large spectrum of pathology. The median (range) proteoglycan score was 1 (0–3), representing superficial and central meniscal loss. The median (range) of DIPEN, NITEGE, and C1,2C scores was 1 (0–3), revealing immunostaining of the femoral and tibial surfaces. The proteoglycan scores exhibited significant positive associations with both histologic evaluation (*p* = 0.03) and DIPEN scores (*p* = 0.02). Additionally, a robust positive association (*p* = 0.007) was observed between the two aggrecanolysis indicators, NITEGE and DIPEN scores. A negative association (*p* = 0.008) was identified between NITEGE and histological scores. The C1,2C scores were not associated with any other scores. Picrosirius red and second harmonic generation microscopy (SHGM) illustrated the loss of the collagen matrix and structure centrally. Proteoglycan and collagen degradation commonly occur superficially in menisci and less frequently centrally. The identification of central meniscal proteoglycan and collagen degradation provides novel insight into central meniscal degeneration. However, further research is needed to elucidate the etiology and sequence of degradative events.

## 1. Introduction

The meniscus plays a crucial role in stabilizing and distributing loads throughout the femorotibial joint compartments. Meniscal degeneration and tears disrupt this function, leading to diminished joint performance and osteoarthritis (OA) [1]. OA is a degenerative disease of the joint organ characterized by progressive articular cartilage fibrillation and erosion, the formation of periarticular and central subchondral osteophytes and sclerosis accompanied by inflammation, fibrosis, and pain. The role of the meniscus in OA pathology has been reviewed by Englund et al. [2]. Factors such as knee misalignment, obesity, and excessive strain from occupational activities or injury can lead to meniscal damage and tears, impairing its function, and are risk factors for OA [2]. Articular cartilage loss occurs at areas of meniscal damage, indicating a strong cause-and-effect relationship between the two. An MRI study has revealed that the prevalence of meniscal damage, including meniscal tear, maceration, or destruction, in the general population (mean age 62) is 35% [3]. Meniscal tears were observed in 31% of people not selected on the basis of having knee or other joint problems, with most (77%) considered degenerative horizontal and complex (40% and 37%, respectively) tears [3,4], and the remaining principally traumatic. The incidence of meniscal damage escalates with age, surpassing 50% in individuals aged 70–90 [3]. MRI meniscal intrasubstance abnormality, within the meniscal core, is linear signals confined within the meniscus thought to represent areas of meniscal degeneration or intrasubstance tears [5].

Quantitative 3T MRI T1ρ and T2 measurements enable differentiation between normal menisci and those with intrasubstance abnormalities or tears [6]. Moreover, MRI meniscal T2 relaxation times also increase with the severity of OA [7]. A meta-analysis has corroborated these findings, highlighting intrasubstance meniscal change as a notable predisposing factor for tears, particularly in patients who experience weight gain [8]. Complementing these findings, an ex vivo study involving human meniscal samples, coupled with histopathological analysis, revealed elevated ultrashort echo time T2 values in both degenerated and torn menisci but not in healthy control tissue [9].

Although MRI meniscal intrasubstance abnormality suggests a gradual degeneration of its extracellular matrix (ECM), the pathophysiology driving it is largely unexplored and is a major knowledge gap in the field.

Collagen is the primary structural molecule of the meniscal ECM [10] and imparts both tensile and compressive properties. Type I collagen predominates [11], with type II collagen present in smaller quantities localized around the meniscal tie fibres [12]. Although biochemical assessments have been conducted to evaluate meniscal collagen content [13,14], and gene expression analyses have explored collagen synthesis [13,14,15,16], the examination of meniscal collagen degradation via the detection of collagenase-generated cleavage products in situ within naturally occurring disease settings has not, to our knowledge, been undertaken in any species. The immunoantibody C1,2C targets a specific collagenase cleavage site neoepitope present in both type I and II collagen. Widely utilized for studying cartilage collagen breakdown in OA across various species [17], these antibodies hold promise for investigating meniscal tissue as well.

Additional important meniscal ECM molecules identified in canine, ovine, and human menisci include the proteoglycans aggrecan, biglycan, perlecan, and decorin [18,19,20]. Aggrecan molecules resist compressive loads and confer meniscal elastic properties [20]. Aggrecan degradation of articular cartilage ECM has been studied extensively and is mediated by specific members of the metalloproteases family (MMPs) and a disintegrin and metalloproteinase with thrombospondin motifs (ADAMTS) [21].

Briefly, proteolysis of the aggrecan core protein occurs at the interglobular domain, yielding specific cleavage sites with neoepitopes that can be detected immunohistochemically using antibodies DIPEN and NITEGE, which are produced by MMPs and ADAMTS, respectively (as reviewed by Roughley and Mort [22]). An experimental investigation into cytokine-induced meniscal degradation in sheep revealed greater levels of ADAMTS-mediated aggrecan cleavage (NITEGE) in the inner meniscus, while MMP-driven aggrecanolysis (DIPEN) predominated in the outer meniscus, as evidenced by Western blot analysis [15].

The horse has become a well-established large animal model for studying spontaneous post-traumatic OA [23], experimental joint disease, and therapeutic approaches for cartilage repair [24]. Our previous investigations into naturally occurring meniscal disease in horses have unveiled parallels with human pathology: meniscal tears and lesions are distributed throughout all meniscal regions [25,26], with the medial meniscus exhibiting the highest frequency of involvement, and the prevalence of disease increases with age [26].

A positive correlation was also found between the meniscal pathology and the presence of OA in the joint [26]. Collectively, these numerous parallels imply that research on equine, non-experimental, spontaneous meniscal pathology could offer valuable insights into human disease. We now wish to expand these findings and explore meniscal ECM molecular degradation by utilizing immunoantibodies to identify specific proteolysis footprints in naturally occurring equine meniscal disease and compare them with the gold standard, site-matched histological analysis of degradation.

The aim of our study encompassed two primary objectives: (1) to examine the degradation of equine meniscal extracellular matrix (ECM) proteoglycan and collagen, employing specific antibodies targeting their cleavage sites, and to correlate these findings with site-matched histological analyses from control and naturally occurring disease meniscal tissues; and (2) to characterize the ECM collagen structure within normal meniscal tissue and compare it with sites exhibiting ECM degradation.

## 2. Results

### 2.1. Meniscal Macroscopic Assessment 

Previous reports have detailed the macroscopic and histologic lesions of the menisci [26]. The median (range) macroscopic ROI meniscal lesion score of the specimens was 4 (1–6), providing control menisci and specimens with a spectrum of disease for further detailed investigation for the present study. Information regarding the investigated specimens can be found in Appendix A.

### 2.2. Quality Control Step

From the 34 selected menisci, 104 ROI blocks were sectioned. Subsequently, only ROI blocks that provided enough sections, with minimal sectioning artefacts (see Section 4.6), that allowed for comparisons between parameters were included in the final analysis. A total of 265 quality slides with minimal artefacts were available from 42 ROI blocks that permitted site-matched histological and immunohistochemistry assessment and comparisons. The complete dataset for the included site-matched histological and ECM parameter ROI scores is provided in Table 1.

### 2.3. Meniscal Histologic Assessment

A total of 38 meniscal ROI sections were included in the histological analysis. Four sections (39–42) of Table 1 were excluded because of suboptimal quality for scoring purposes. The median (range) total ROI histological score was 3 (0–9), providing a large spectrum of pathology for study (Figure 1, Figure 2 and Figure 3). Histologic lesions of the meniscal femoral and tibial surfaces included mild to moderate fibrillation, occasionally extending deeper into the meniscal substance. Inner border lesions were prevalent and included some with severe disruption and tissue loss (Figure 2 and Figure 3). The distribution of scores is provided in Appendix A.

### 2.4. Meniscal ECM Assessment

#### SOFG Assessment

A total of 40 ROI sections were included in the SOFG analysis. The inter-rater agreement for assessing SOFG staining scores was substantial (0.7). Scores reported by JD were utilized. The median (range) ROI SOFG score was 1 (0–3). The frequencies of scores are reported in Figure 4. A homogeneous SOFG stain (score 0) throughout the section was detected in 13% of sections (5/40). Superficial loss of SOFG stain on the femoral and tibial surfaces (score 1), indicating reduced proteoglycan content, occurred in 43% (17/40) (Figure 4). The pattern of both superficial and central meniscal core SOFG reduction (score 2) was the most frequently observed, in 43% (17/40). Generalized loss of SOFG (score 3) was observed in only one section.

### 2.5. Meniscal ECM Proteoglycan Degradation 

The frequencies of all immunostain scores are reported in Figure 4, and examples are provided in Figure 1, Figure 2 and Figure 3.

#### 2.5.1. NITEGE Immunostaining

A total of 42 ROI sections were included in the NITEGE analysis. The inter-rater agreement for NITEGE was very good at 0.8. The median (range) ROI score for the NITEGE immunostain was 1 (0–3). Absence of NITEGE staining (score 0) was observed in 12% (5/42) of ROIs. NITEGE immunostaining on the femoral and tibial surfaces (score 1), indicative of proteoglycan degradation, occurred in 43% (18/42). Both superficial and central meniscal core immunostaining with NITEGE (score 2), indicating spontaneous intrasubstance proteoglycan degradation, was observed in 31% (13/42). Generalized NITEGE staining (score 3) was identified in 14% (6/42) of ROIs.

#### 2.5.2. DIPEN Immunostaining

A total of 39 ROI sections were included in the NITEGE analysis. The inter-rater agreement for DIPEN on these sections was excellent at 0.9. The median (range) ROI score for DIPEN was 1 (0–3). No DIPEN staining (score 0) was observed in 36% (14/39). DIPEN immunostaining on the femoral and tibial surfaces (score 1) occurred in 23%. (9/39). Meniscal core immunostaining with DIPEN, indicative of central proteoglycan degradation (score 2), occurred in 31% (12/39). Generalized DIPEN staining (score 3) was identified in 10% (4/39). Colocalization assessment was not possible in a limited number of ROIs, as occasional poor-quality sections were eliminated from the analysis (Table 1). Colocalization of NITEGE and DIPEN staining was observed in the majority of specimens where it was assessed (64%; 25/39) (Table 1).

### 2.6. Meniscal Collagen Degradation: C1,2C Immunohistochemistry

A total of 40 ROI sections were included in the analysis. The inter-rater agreement for C1,2C immunostaining was substantial at 0.7. The median (range) C1,2C score was 1 (0–3). A lack of focal staining (score 0) was observed in 10% (4/40). Enhanced C1,2C immunostaining of the femoral and tibial surfaces (score 1) was present in 55% (22/40). Superficial and meniscal core immunostaining with C1,2C, indicative of central intrasubstance collagen degradation (score 2), was observed in 33% (13/40). Generalized C1,2C staining throughout the meniscus ECM (score 3) was identified in only one section.

### 2.7. Meniscal Collagen Structure: Picrosirius Red Polarized Light Microscopy

Although the well-organized structure of healthy menisci was easily identifiable, it was determined that a reliable, repeatable scoring method was not possible to capture the degradation of the collagen. Consequently, the description that follows is qualitative. Forty-two sections were selected for this analysis and included ROIs with minimal and advanced histological lesions. Healthy menisci exhibited a highly organized collagen structure. A thin lamellar layer of collagen fibres was discernible, running parallel to both the femoral and tibial surfaces and extending from the capsular attachment at the outer aspect of the menisci (Figure 5a–c). Large tie fibres radiated from the outer meniscal surface centrally. Tie fibres appeared more abundant in the tissue sections from the caudal horns compared to the meniscal body and cranial horns.

Some menisci exhibited evident loss of collagen organization on the surfaces and at the inner border. Although the tie fibres extending from the outer meniscal border and the bundles from the lamellar surfaces were less defined, they were still present (Figure 5g–i). Additionally, a decrease in collagen density in the meniscal core was observed in some sections, consistent with an intrasubstance degenerative process (Figure 5). In a specimen sourced from a clinically diagnosed OA compartment, an open lacey pattern with small holes was observed when compared to age-matched specimens (Figure 5).

### 2.8. Meniscal Collagen Structure: SHGM

A subset of 20 ROI sections was selected for SHGM ultrastructural analysis. In control meniscal ROIs, collagen fibres covering the femoral and tibial contact surfaces exhibited a radial orientation and ran parallel to each other from the outer meniscal region to the inner border. The tie fibres maintained a radial orientation across their entire length. However, in specimens with lesions of the inner border, the alignment of collagen fibres was either lost or disrupted. Additionally, a lacey appearance was evident in degenerated menisci, indicating decreased collagen density (Figure 6).

### 2.9. Associations between Site-Matched Meniscal Histological and ECM Scores

The results of ECM variables were compared with histological assessment to explore associations of the degradation patterns with various stages of meniscal disease. The ROI proteoglycan loss SOFG scores exhibited significant positive associations with both histologic and proteoglycan degradation (DIPEN) scores (*p* = 0.03 and *p* = 0.02, respectively). A very strong positive association (*p* = 0.007) was also found between the two aggrecanolysis (NITEGE and DIPEN) scores. However, a negative association (*p* = 0.008) was identified between NITEGE and histological scores. Interestingly, the meniscal ROI collagen degradation (C1,2C) scores were not associated with any other scores.

## 3. Discussion

This study offers novel insights into the molecular breakdown of the principal meniscal ECM molecules, aggrecan and collagen, in naturally occurring meniscal pathology by employing immunoreactive antibodies targeting specific neoepitopes at their cleavage sites. They identify unique footprints of aggrecanolytic (DIPEN-MMPs; NITEGE-ADAMTS) and collagenase (C1,2C) enzymes [27] in the meniscal ECM. A significant association was found between the MMP degradation of proteoglycan, employing the DIPEN antibody, and site-matched focal histological degeneration in the meniscal ROIs examined. Although no similar association was identified with the C1,2C immunoantibody, both picrosirius red staining and SHGM analysis revealed the structural disruption and occasional absence of the intricate, highly organized collagen network crucial for the normal biomechanical function of the meniscus in tissue with histological evidence of disease. In addition, in specific ROI sections, these proteolytic events also occasionally occurred independently of identified histological changes, suggesting they may also signify early molecular events preceding macroscopic structural damage. However, additional investigation is necessary to validate this hypothesis.

A consistent trend of diminished proteoglycan content, as evidenced by SOFG staining, was frequently observed at the surfaces of the menisci. This finding, indicating reduced proteoglycan content, corresponds with earlier studies conducted on porcine and bovine menisci [28,29]. We postulate that this meniscal surface alteration is a component of normal meniscal turnover. The association between the SOFG score and the histological degradation score suggests a link between the degradation of ECM aggrecan and meniscal degeneration. However, findings regarding proteoglycan content in diseased menisci have been inconsistent. For instance, LeGraverand et al. [30] noted that intrameniscal tears were encircled by regions of proteoglycan-deficient ECM within the central region, whereas a tissue abundant in proteoglycans was observed in the outer portion of lapine menisci in experimental OA. In humans with OA, increased meniscal glycosaminoglycan content has been reported [31,32], a trend similarly observed in experimental OA models in rabbits [33]. Sun et al. [29] also noted an increase in SOFG staining primarily within the deep meniscal zone of human OA menisci, suggesting an anabolic response. Levillain et al. [33], employing biphotonic confocal microscopy and histology in a rabbit OA model, observed a more pronounced proteoglycan staining in meniscal regions characterized by disorganized, less aligned, and undulated collagen fibres. Combined these conflicting findings with respect to meniscal proteoglycan content may reflect different study conditions, natural disease, duration of disease or experimental methods, or species differences.

A positive association between the meniscal ROI SOFG proteoglycan loss score and its MRI cleavage identified by DIPEN immunostaining, in addition to the histological score, suggests that MMP has a role in meniscal ECM degeneration. However, the SOFG loss score was not associated with the ADAMTS-mediated degradation of aggrecan detected by NITEGE. Additionally, high NITEGE and DIPEN scores of 3 were occasionally observed in certain meniscal sections with low histological scores. These observations suggest that aggrecanolysis might represent an early event in the progression of meniscal disease.

The negative associations between NITEGE immunostaining and histological scores corroborate the findings of others [34] who observed a reduction in extracellular matrix NITEGE in human meniscal sections with advanced degeneration and speculated that it was due to loss of the epitope fragments of the molecule aggrecan with increased matrix degeneration [34]. In vitro experiments with bovine meniscal explants following exposure to IL-1 have also shown that aggrecanases have a key role in meniscal degeneration, in addition to being part of normal aggrecan metabolism in the meniscus [35].

The collagen network organization in adult equine menisci from the control meniscal ROIs was similar to that previously observed in humans [36,37] and other animal species [14,28,29,38]. Collagen fibres were more dense and aligned parallel at the femoral and tibial surfaces. The radial tie fibres branched out centrally from the outer border in conjunction with smaller diameter tie fibres also extending from the femoral and tibial surfaces. Picrosirius red stained sections and SHGM illustrated destruction, and sometimes loss, of the highly ordered, complex collagen network, essential for the normal biomechanical function of the meniscus in some sections. The collagen bundles were less compact and undulated, similar to descriptions from human reports [13,31,33].

C1,2C staining, which identifies specific collagenase cleavage sites of ECM type I or II collagen, was most commonly observed, superficially, over both meniscal surfaces. The collagen degradation scores were not associated with the ROI histological scores. One possible interpretation is that the upstream collagen molecular degradation events arise before overt structural changes manifest or, alternatively, at a tipping point in late-stage disease alone. Despite the identification in the meniscal ROIs of intrasubstance proteoglycan loss and degradation and collagen degradation, no association was found between the scores of these two components of the meniscal ECM. This suggests that, at least in the initial stages of degeneration, collagenolytic and aggrecanolytic events are not closely linked or that they have different inciting factors. It is worth noting that the chronology of these proteolytic degradative events in OA cartilage remains a matter of debate, with some proposing that aggrecanolytic events occur first [39,40,41,42,43], while others argue that collagenolytic events also occur early [17,44]. Regarding meniscal collagen network degradation, it has been theorized that it would occur in areas with significant proteoglycan loss [15] as aggrecan would shield the collagen network until a sufficient amount of aggrecan was released, allowing MMPs to access the collagen molecules to initiate collagenolysis [42]. A recent human meniscus degeneration ex vivo explant model found that though cytokine Il-1 induced glycosaminoglycan release, combinations of oncostatin M and TNF alpha induced a much stronger catabolic effect, and the authors propose that these molecules trigger meniscal ECM degradation [45].

These observations confirm that collagen cleavage by collagenases is a component of progressive degeneration of the meniscus. Although we postulate that this loss of tissue would compromise optimal meniscal function, it has been speculated by others that menisci with a decrease in the number of compact collagen bundles can still resist hoop stresses [33]. However, we suspect that this process may progress and be a prelude to tears because of a weak ECM unable to sustain either physiological or supraphysiological forces.

It is acknowledged that this investigation has several limitations. A limited number of ROIs were available for study, as it proved challenging to have multiple, site-matched high-quality sections within the same ROIs to permit analysis across all the parameters Others have also reported the challenge of cutting menisci for intact sections [45]. Although the stifle joints of origin were examined to identify cartilage lesions, their exact clinical status was unknown, except for one specimen that had clinical OA and severe meniscal damage. Furthermore, some of the meniscal ROIs were harvested from joints with OA lesions, and consequently, the observations here most likely represent degenerative meniscal lesions rather than lesions associated with acute traumatic events. At the same time, this could be considered an advantage as most meniscal tears are of a degenerative type [3]. The association of aging and inflammatory processes like OA with meniscal degeneration and the contribution of each of these factors to degradation remains to be elucidated [45].

An important strength of the current report is the investigation of naturally occurring meniscal disease tissue, including from older animals, in contrast to studies conducted on experimental animals, as it provides a more accurate reflection of real-life conditions within the joint environment. We speculate that the proteolysis that we observed also arises in people with MRI meniscal intrasubstance degradation, but further studies employing MRI combined with the ECM assessment we describe here in human meniscal specimens will be necessary to confirm this.

In summary, proteoglycan loss and collagen and proteoglycan degradation commonly occur superficially on meniscal surfaces and less frequently centrally. The meniscal intrasubstance collagen degradation and proteoglycan loss provide evidence for central meniscal degeneration caused by proteolysis, similar to articular cartilage, as associated collagen structural alterations were observed on some sections co-localized to these sites. Further research is now needed to determine the etiology and sequence of degradative events affecting the menisci as well as the primary site where these take place.

## 4. Materials and Methods

### 4.1. Source of Meniscal Tissue

Meniscal samples were obtained from an equine stifle tissue bank in accordance with the protocol approved by the University of Montreal’s Institutional Animal Care and Use Committee (IACUC). One stifle joint from adult horses (*n* = 17) was harvested for the tissue bank and included various breeds, sexes, and ages spanning from 3 to 27 years. Information regarding signalment and the origin of the banked stifle joints is provided in Appendix A. The joints were evaluated macroscopically immediately, as described previously [26,46], or stored in saline-soaked gauze in sealed plastic bags to avoid desiccation and frozen at −20 °C until subsequent analysis. All the articular surfaces were grossly assessed for cartilage changes for the archive records (criteria for macroscopic assessment are provided in Appendix A). Sketches were created, and high-resolution photographs were also taken. The tissue bank included stifle joints with minimal changes (controls) and a spectrum of degenerative OA.

For the present study, menisci (*n* = 34) and associated archived information were retrieved from the tissue bank. Some of the specimens were part of a prior study which documented the distribution and types of meniscal lesions in equine joints [26]. Macroscopic evaluation of the meniscal tibial and femoral surfaces of the cranial horn, body, and caudal horn regions (ROIs) was also conducted for the current study by two evaluators, who reached a consensus. The criteria for the ROI macroscopic scores are in Appendix A.

The macroscopic cartilage degradation scores for the medial and lateral femorotibial joint compartments (femoral condyle and tibial plateau) were also retrieved from the records to document the health status of the meniscal joint compartment of origin for the present study (Appendix A).

### 4.2. Processing of Meniscal Tissue

The menisci were thawed in water. Tissue blocks were cut from the centres of the ROIs and fixed in 10% formalin for 2 h, and then decalcified in 20% EDTA for 2 weeks and embedded in paraffin, as described [26]. Prior to sectioning, the ROI paraffin blocks were also treated with a decalcification solution (Surgipath Decalcifier II, Leica Biosystems, Richmond, Illinois, USA) that facilitated sectioning. For the present study, multiple (*n* = 7) serial, site-matched sections, 5 μm in thickness, were cut from the central portion of each ROI using a microtome (Figure 7) for histological stains and immunohistochemistry to allow comparison across the selected ECM parameters. Digitalization of all sections was performed following the staining procedures utilizing a LeicaDM 4000B microscope (Leica Biosystems, Richmond, IL, USA) and Panoptiq™ v.1.4.3 (ViewsIQ, Richmond, BC, Canada) computer software for tissue bank archives.

### 4.3. Histological Assessment

Sections from each ROI were stained with hematoxylin–eosin–phloxine and saffron (HEPS) and scored based on established criteria provided in Figure 8 [26]. Briefly, the femoral and tibial surfaces, as well as the inner border of each ROI, were assessed for structural changes by a board-certified veterinary pathologist. The scores were summed to generate a meniscal ROI histological score, ranging from 0 to 9, for subsequent comparisons with ECM molecular degradation scores.

### 4.4. Meniscal ECM assessment

#### 4.4.1. Proteoglycan Aggrecan Staining with Safranin O Fast Green (SOFG)

Site-matched sections corresponding to histological ROI sections were cut and stained with SOFG to identify ECM proteoglycan loss. Initially, proteoglycan staining patterns were evaluated by three observers to establish a scoring atlas via consensus agreement. The criteria for ROI SOFG-stained section scores are outlined in Figure 8. The SOFG-stained sections were independently assessed by two blinded evaluators utilizing the established atlas to provide an SOFG score for each ROI.

#### 4.4.2. Immunohistochemistry for Aggrecanolysis (NITEGE and DIPEN) and for Collagen Cleavage: C1,2C (Col 2 3/4Cshort)

ROI sections were incubated with rabbit antibodies (NITEGE or DIPEN) that target the proteoglycan aggrecan cleavage sites [21]. Meniscal collagen degradation was detected using a C1,2C (Col 2 3/4Cshort) polyclonal rabbit antibody, which identifies a cleavage neoepitope shared by type I and II collagens and generated by collagenases [39]. Further details regarding the immunohistochemical techniques can be found in Appendix B. The ECM molecular cleavage patterns, revealed by the specific antibodies, were then scored. An atlas was developed for scoring purposes through consensus agreement of two observers. The criteria for the meniscal ROI ECM aggrecanolysis and collagen cleavage are presented in Figure 8. Meniscal ECM degradation was independently scored by two blinded individuals to provide a NITEGE or DIPEN and Col 2 3/4Cshort proteolytic score for each ROI.

### 4.5. Quality Control Step for Histological and Immunohistochemistry-Stained Sections for Inclusion in the Analysis and Comparisons

Obtaining intact, high-quality, site-matched sections of the adult meniscus with minimal sectioning artefacts or tissue damage, particularly in older specimens, is a challenge due to the meniscus’s complex, dense, and occasionally mineralized collagen structure. This leads to sectioning artefacts and undesirable variability in the samples being studied. This challenge has also been documented by other researchers in the field [45]. Every possible measure was undertaken to obtain quality sections. A technician with over 20 years of experience sectioning cartilage and decalcified bone tissue cut the meniscal paraffin blocks with a microtome (Thermo Scientific, Waltham, MA, USA, Microm HM 355S). An initial quality control measure was also implemented to determine the suitability of each section for assessment. Repeated sectioning was performed to improve quality. ROIs displaying suboptimal section quality (artefactual tearing or loss of tissue) were consequently excluded from the study. Only ROIs that had enough high-quality, site-matched sections that permitted assessment and comparisons across the parameters were selected. The quantity of ROI sections included in the final analysis is detailed for each parameter.

### 4.6. Meniscal Collagen Structure: Picrosirius Red Polarized Light Microscopy and SHGM 

Picrosirius red-stained sections were also analyzed for collagen structure using polarized light microscopy [47]. Additionally, a subset of ROI sections (*n* = 20) was chosen for SHGM to qualitatively investigate collagen ultrastructure in areas of histologically healthy-appearing meniscal tissue and lesions ranging from surface fibrillation to severe intrasubstance degeneration. The methods employed were adapted from those previously described for cartilage [47,48] (Appendix B).

### 4.7. Statistical Analyses

The inter-observer agreement for histological, SOFG, NITEGE, DIPEN, and C1,2C ROI scores was assessed with a weighted Kappa test. Cochran–Mantel–Haenszel tests were employed to explore the relationship in ROIs between meniscal structural degeneration (histological scores) and ECM degradation (SOFG, NITEGE, DIPEN, and C1,2C scores) from site-matched sections. A level of *p* < 0.05 was considered statistically significant (SAS v. 9.4 (Cary, NC, USA) software).

## Figures and Tables

**Figure 1 ijms-25-06439-f001:**
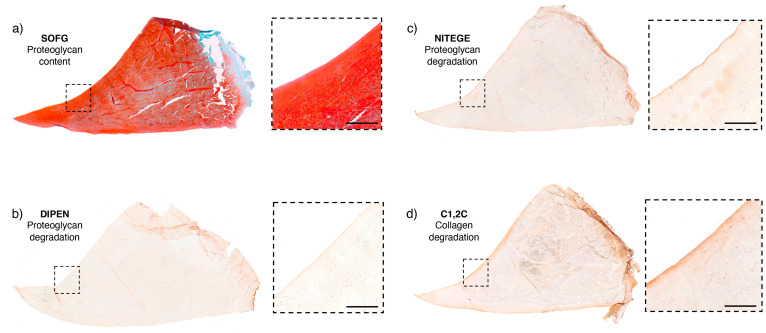
Site-matched ROI sections from a healthy meniscus stained with SOFG, NITEGE, DIPEN, and C1,2C. These sections are from a 5-year-old horse. They represent healthy tissue as the HEPS histological score was 0, the lowest possible, revealing an intact structure. (**a**) The SOFG-stained section reveals an intact meniscal structure with uniform uptake of the stain but slightly paler in the centre. The splits are artefacts. The broken box corresponds to the magnification on the right, revealing a smooth, intact femoral surface. The SOFG score was 0. (**b**) The DIPEN score was also 0, revealing no immunostaining. (**c**,**d**) Both the NITEGE and C1,2C were attributed a score of 1 as the magnified images reveal some uptake of the immunostain on the femoral surface. Key: SOFG—safranin O fast green scale bar: 500 µm.

**Figure 2 ijms-25-06439-f002:**
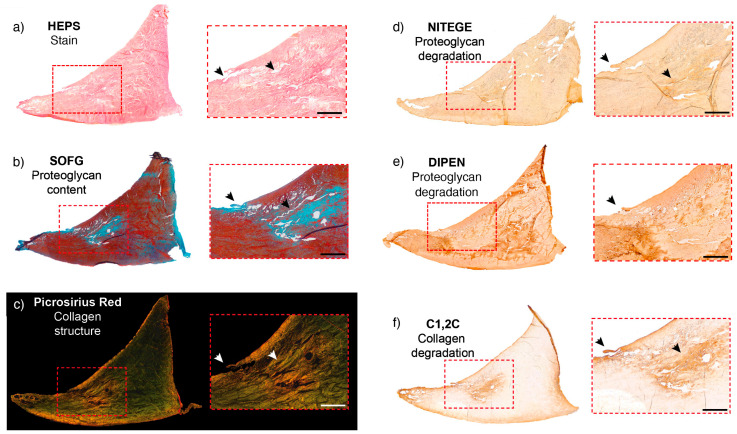
Site-matched meniscal ROI sections revealing central or intrasubstance extracellular matrix changes, including proteoglycan loss and focal degradation of proteoglycan and collagen molecules combined with disruption of collagen structure. The meniscal ROI is from a 26-year-old horse with a partial tear in the meniscus. The dashed rectangles in the right panel of the image illustrate sites of corresponding magnified inserts. Arrowheads highlight surface and central anomalies. (**a**) HEPS-stained sections. (score 6) This section reveals disruption of structure that includes an abnormal structure with loss at the inner border, a partial tear from the femoral surface, disorganized tissue centrally, and loss of tissue on the tibial surface. (**b**) The SOFG-stained section reveals a focal pale blue area centrally, corresponding to proteoglycan loss and also at the surface (Score 2). There appears to be a loss of tissue structure also at this site on the magnified insert to the right. (**c**) The corresponding site-matched picrosirius-stained section and magnified insert clearly illustrate the loss of the organized collagen structure within the lesion and also the inner border changes observed in (**a**). (**d**) A site-matched section to (**a**) where the NITEGE antibody has a background generalized uptake with a focal enhanced uptake centrally (Score 2) revealing increased central degradation of proteoglycan molecules. (**e**) DIPEN-stained section score 3. (**f**) A site-matched C1,2C-stained section to (**a**) shows a clearly demarcated enhanced uptake centrally but also peripherally (Score 2). This section appears to have a loss of collagen at the inner border in addition to the central lesion. Scale bar = 1 mm.

**Figure 3 ijms-25-06439-f003:**
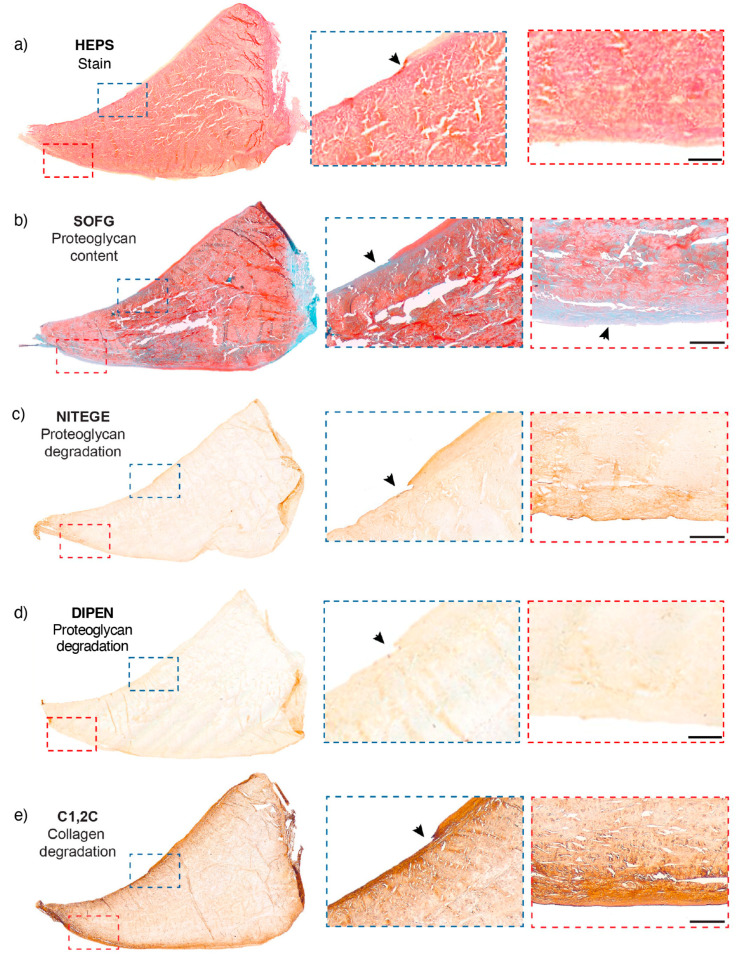
Site-matched meniscal ROI sections from a meniscus of a 10-year-old horse. (**a**) HEPS-stained section. Loss of tissue at the inner border is evident. An erosion of tissue on the femoral surface is evident (arrowhead in magnified image). The ROI histological score was 8. (**b**) SOFG-stained section revealing a focal loss of proteoglycan content at the femoral and tibial surfaces (Score 1). The disrupted architecture in the middle is a processing artefact as meniscal tissue is challenging to section. The dashed rectangles to the right of the image illustrate sites of corresponding magnified inserts and arrowheads are pointing at the identified lack of proteoglycan (decreased red stain). (**c**) A section immunostained with NITEGE, a degradation product of proteoglycan (Score 1). There is increased uptake at the femoral and tibial surfaces when compared to similar regions within the same section. The magnified images on the right illustrate the enhanced uptake of the immunostain. (**d**) A section toimmunostained with DIPEN (score 0). No staining is evident. (**e**) A section an increased uptake of the C1,2C antibody for collagen degradation at both the femoral and tibial surfaces and inner border (Score 1), at the level of the previously identified proteoglycan loss in this ROI. Scale bar = 500 µm.

**Figure 4 ijms-25-06439-f004:**
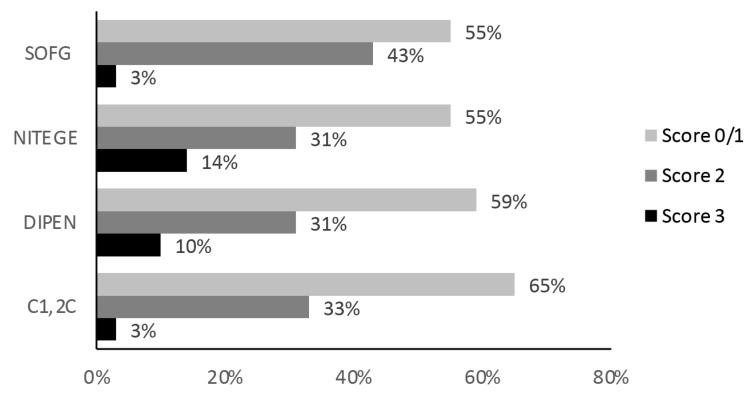
The score frequencies for each of the parameters investigated in the ROIs.

**Figure 5 ijms-25-06439-f005:**
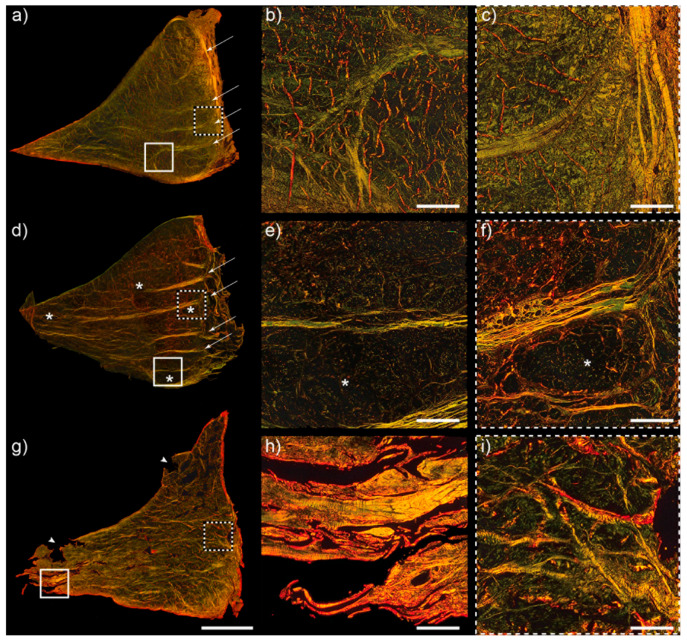
Meniscal collagen structure in health and disease illustrated with picrosirius red stain and polarized light microscopy. The squares in the left-panel ROIs correspond to the magnified inserts in the 2 panels on the right. (**a**–**c**) A cranial horn ROI section of a medial meniscus from a control specimen aged ten years. ROI histological score was 3. Thin, intact lamellar layers are evident on both femoral and tibial surfaces. Tie fibres arborize from the outer meniscus (white arrows). (**d**–**f**) ROI from the body of a medial meniscus from a 14-year-old horse. ROI histological score was 2. Evident decrease in collagen density is illustrated by an asterisk. The tie fibres in this ROI section are of larger diameter than in specimen (**a**). This horse had clinical osteoarthritis in the joint. (**g**–**i**) The cranial horn ROI of a medial meniscus from a 27-year-old horse. ROI histological score was 7. There is a generalized loss of collagen organization and loss of meniscal integrity at the level of the inner border and femoral surface. Arrowheads highlight tears and clear tissue loss. Scale bar, (**a**,**d**,**g**) = 4 mm. Scale bar, others = 500 µm.

**Figure 6 ijms-25-06439-f006:**
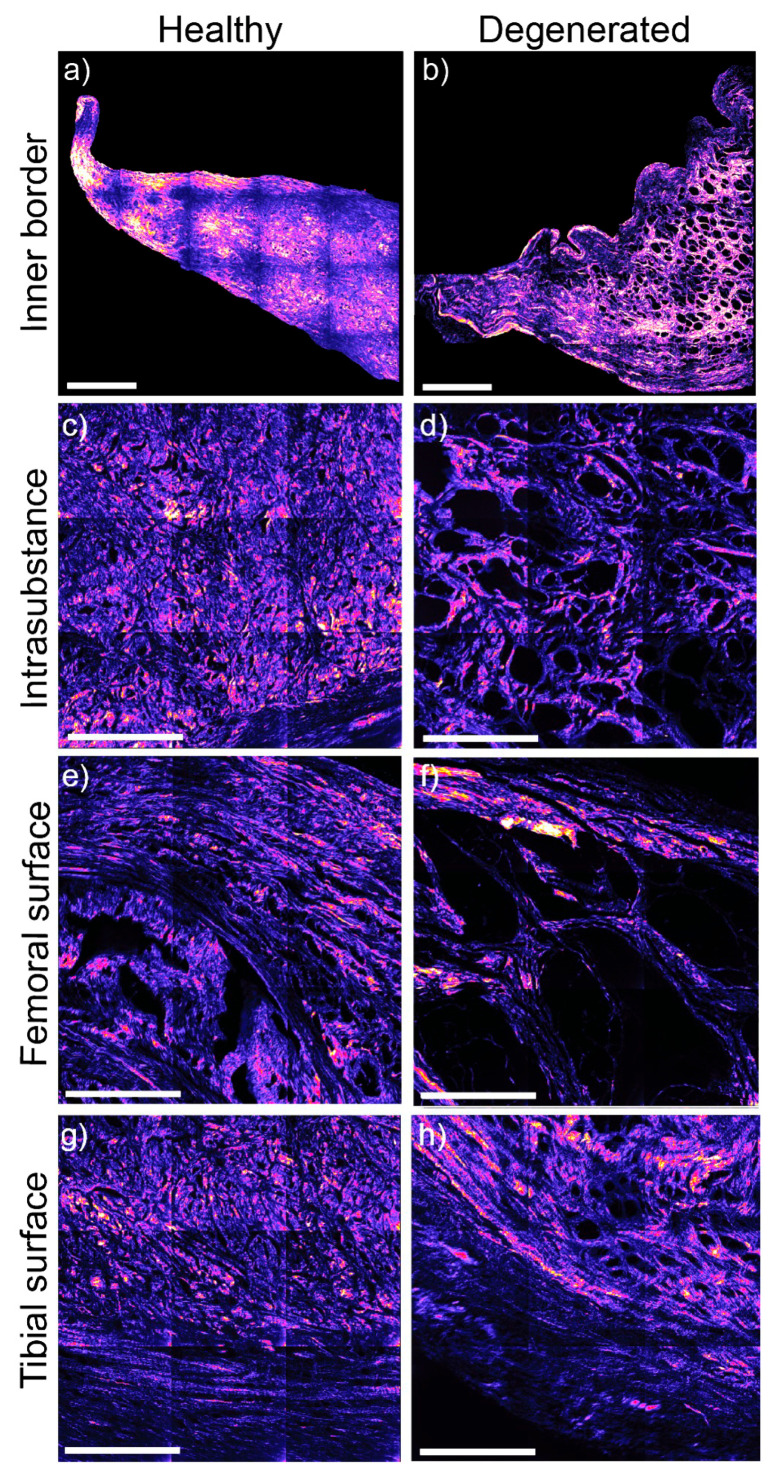
Equine meniscal ROI collagen ultrastructure with second harmonic generation microscopy (SHGM) in healthy and degenerated specimens. The samples are imaged from key meniscal sites (inner border, intrasubstance, and both the femoral and tibial surfaces) from a control section from an 8-year-old horse and with site-matched degenerate meniscal tissue from a 23-year-old horse. The healthy control collagen structure is compact and organized (**a**,**c**,**e**,**g**) when compared with the degenerate samples on the right (**b**,**d**,**f**,**h**). In the samples with disease, there is an evident loss of collagen ultrastructure, and small (**b**,**d**,**h**) to larger holes (**f**) that give it a lacey appearance (**b**,**d**,**f**) and represent a loss of collagen molecules from the tissue. Scale bars of images (**a**,**b**) = 200 µm. Scale bars for images (**c**–**h**) = 300 µm.

**Figure 7 ijms-25-06439-f007:**
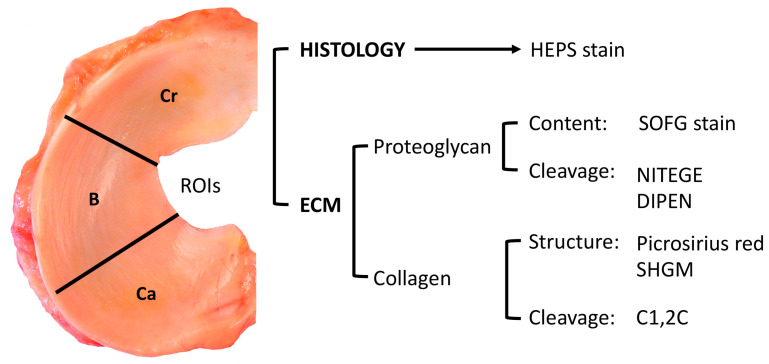
Study design. Meniscal specimens from one stifle per horse were retrieved from the tissue bank (*n* = 34). They were divided into 3 ROIs: cranial-Cr and caudal-Ca horns and body-B. Seven site-matched sections were cut from the centre of each ROI using a microtome for histological stains and immunohistochemical analysis to allow comparative analysis. NITEGE, DIPEN, and C1,2C are antibodies. Key: ECM, extracellular matrix; HEPS, hematoxylin, eosin, phloxine, and saffron stain; SOFG, safranin O and fast green stain; SHGM, second harmonic generation microscopy.

**Figure 8 ijms-25-06439-f008:**
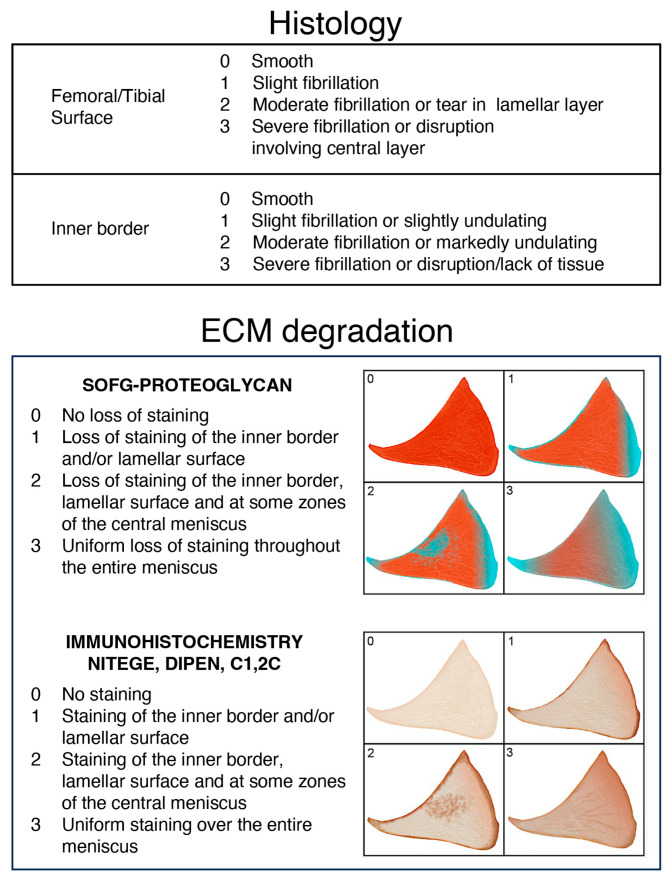
Criteria for the meniscal ROI histological and extracellular matrix scores. The histological score was the same as employed previously [26]. The femoral and tibial surfaces and inner border were all scored for structural changes employing a score from 0 to 3. The extracellular matrix proteoglycan content (SOFG) score (0–3) was based on the pattern and extent of the SOFG staining within the section. The immunohistochemical scores for proteoglycan (NITEGE and DIPEN) degradation and for collagen (C1,2C) ranged from 0 to 3. The accompanying atlas schema illustrates the score patterns.

**Table 1 ijms-25-06439-t001:** The meniscal ROIs included in the final analysis, following the quality control step (Section 2.2) and the corresponding scores for each of the parameters assessed in site-matched sections. Following the quality control step, the listed ROIs were judged to have ample quality sections, with minimal artefacts to permit comparisons across all modalities, “-“ indicates that scoring was not possible in one section. The ROI was included as information was available for other parameters.

ROI Identification	Horse No.	Meniscus	ROI Region	Histological HEPS Score	Proteoglycan Score (SOFG)	Proteoglycan Degradation (NITEGE)	Proteoglycan Degradation (DIPEN)	Meniscal Collagen Degradation (C12C)
1	5	L	Cr	0	0	1	0	1
2	7	L	Ca	0	1	3	2	1
3	8	L	Ca	0	1	2	1	1
4	10	L	Cr	0	0	2	2	2
5	10	L	B	0	2	2	0	2
6	10	M	Cr	0	1	2	1	2
7	15	L	B	0	-	0	2	-
8	15	L	Ca	0	1	1	3	1
9	16	L	Cr	0	-	2	2	-
10	16	L	Ca	0	1	3	3	2
11	17	L	Cr	0	2	3	2	1
12	17	L	B	1	2	1	2	0
13	10	M	Ca	2	1	1	0	2
14	11	L	Cr	2	0	2	2	1
15	16	L	B	2	2	1	-	1
16	16	M	Cr	2	2	1	2	1
17	17	M	Cr	2	2	1	0	0
18	17	M	B	2	1	0	0	0
19	17	M	Ca	2	0	0	0	0
20	1	L	B	3	2	1	1	1
21	4	M	Cr	3	1	2	1	2
22	7	L	B	3	2	1	1	1
23	9	L	Cr	3	2	2	-	1
24	12	M	B	3	1	3	3	1
25	14	L	Ca	3	2	1	1	2
26	6	M	Cr	4	2	3	2	2
27	7	M	B	4	2	1	1	2
28	13	L	Ca	4	1	2	0	1
29	16	M	B	4	3	1	-	1
30	9	M	B	5	2	1	0	2
31	9	M	Ca	6	2	2	2	3
32	12	L	Ca	6	2	2	3	2
33	13	M	Ca	6	1	2	1	1
34	2	L	Cr	7	0	1	0	1
35	3	M	Cr	7	1	2	0	1
36	14	M	Ca	7	2	0	0	1
37	2	M	Ca	8	1	1	0	1
38	3	L	Ca	9	1	0	0	2
39	7	M	Cr	-	1	1	2	1
40	7	M	Ca	-	2	3	2	2
41	10	L	Ca	-	1	1	1	1
42	13	M	B	-	1	1	0	1

## Data Availability

The research data are available from the corresponding author.

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
