# Peer review of "Degradation of Proteoglycans and Collagen in Equine Meniscal Tissues"

_ijms, 2024, doi:10.3390/ijms25126439_

Round 1
Reviewer 1 Report
Comments and Suggestions for Authors
· The authors should use numbers to identify author’s affiliations and not symbols.
· Title should be revised specifying that the authors analyzed equine menisci.
· Lines 21-24: r values should be added for correlations.
· Line 25: SHGM should be defined.
· Line 34: OA should be defined.
· The authors should add a brief description of OA pathology. A better description of meniscal changes occurring in OA pathology should be included (DOI:10.4081/ejh.2019.2998 etc).
· Lines 37-39: in healthy people?
· Lines 40-42: references should be added.
· Line 38: resepctively should be corrected.
· Lines 82-83: could the authors better explain what they mean with equine spontaneous meniscal disease?
· Line 99: table s1: the authors reported that only one horse had OA. What about the others?
· Line 100: full-stop should be deleted.
· Line 101:why did the authors store in saline-soaked gauze and place the tissues at -20°C? The use of saline-soaked gauze could disrupt the tissues.
· Section 2.1: Table s2. What kind of scores did the authors use? They should be added with specific references. Did the authors start from OARSI score or Pauli score etc?
· Line 117: did the authors use frozen tissues?
· Line 119: full-stop is missing.
· Figure 1: the sample called B has two sides that match the samples Cr and Ca. It is not clear to me how it was cut.
· Section 2.5: supplier of all antibodies should be added. Dilutions used of the antibodies should be added. I noticed that the antibodies were a gift. This point should be better explained. Are these home-made or commercial antibodies? If home-made, did the authors verify the specificity?
· Software used for statistical analysis should be added.
· Section 3.2: this part should be better explained. The authors discussed of 42 ROI blocks but there are 34 menisci and each menisci should be divided into three paraffin blocks, for a total of 102 blocks and not 42.
· Line 217, 245, 255: why 38 or 40 or 42? Is this related to table 1?
· Table 1 is not clear. It is not clear which horse each ROI corresponds to.
· Line 255: um should be corrected.
· Lines 328-329: what are these controls?
· Section 3.9: is there a correlation with age of the horses? Could the authors check?
· It is not clear to me why the authors did not divide and compare the samples based on the presence/absence of OA and tears.
· Lines 359-361: this sentence should be tone down as it is unclear how many horses had OA and how many had tears etc.
· I am not able to find appendix A, B and C. Authors should check.
Author Response
Thank you to the Editor and reviewers for time taken to review the manuscript and for the valuable input that has helped improve it.
We apologize as it seems to us that information included in appendices was not perhaps shared with the reviewers on the initial review and may have contributed to some confusion. Supplementary information and appendices are again provided.
Ms. Ref. No.: ijms-2971320
Title: Degradation of Proteoglycans and Collagen in Meniscal Tissue
|
|
Comments and Suggestions for Authors
- The authors should use numbers to identify author’s affiliations and not symbols.
Authors’ response: Thanks for the suggestion
Authors’ action: Numbers have been added to identify the authors' affiliations
- Title should be revised specifying that the authors analyzed equine menisci.
Authors’ response: Thank you
Authors’ action: We have added equine to the title.
- Lines 21-24: r values should be added for correlations.
Authors’ response: Since we used the Cochran-Mantel-Haenszel (CMH) test to examine the association between our ordinal scores, it is not possible to provide a correlation value for each test. The test statistic with CMH is a chi-square value and this cannot be converted to a correlation in the usual sense.
Authors’ action: We have removed mention of the word correlation and now use the word association instead.
- Line 25: SHGM should be defined.
Authors’ response: Thank you for this comment.
Authors’ action: The definition has been added.
- Line 34: OA should be defined.
Authors’ response: Thank you for this comment.
Authors’ action: Osteoarthritis (OA) has been added.
- The authors should add a brief description of OA pathology. A better description of meniscal changes occurring in OA pathology should be included.
Authors’ response: Thank you for the suggestion.
Authors’ action: We have now added the following:
Line 45-53 “OA is a degenerative disease of the joint organ characterized by progressive articular cartilage fibrillation and erosion, the formation of periarticular and central subchondral osteophytes and sclerosis accompanied by inflammation, fibrosis, and pain. The role of the meniscus in OA pathology has been reviewed by Englund et al[2] . Factors such as knee misalignment, obesity, and excessive strain from occupational activities or injury can lead to meniscal damage and tears impairing its function and are risk factors for OA[2]. Articular cartilage loss occurs at areas of meniscal damage, indicating a strong cause-and-effect relationship between the two.”
- Lines 37-39: in healthy people?
Authors’ response and action: We have inserted “people not selected on the basis of having knee or other joint problems” to provide additional clarification.
- Lines 40-42: references should be added.
Authors’ response and action: A reference has been added as requested.
Line 64-66“MRI meniscal intrasubstance abnormality, within the meniscal core, are linear signals confined within the meniscus thought to represent areas of meniscal degeneration or intrasubstance tears.[2]”
Ahmad, R. Intra-substance meniscal changes and their clinical significance: a meta-analysis. Sci Rep 11, 3642 (2021). https://doi.org/10.1038/s41598-021-83181-5
- Line 38: resepctively should be corrected.
Authors’ response and action: Thank you. The correction was made.
- Lines 82-83: could the authors better explain what they mean with equine spontaneous meniscal disease?
Authors’ response: Thank you for this comment.
Authors’ action: We have modified the text as follows to provide additional clarity:
L 120-122 “Collectively, these numerous parallels imply that research on equine, non-experimental, spontaneous meniscal pathology could offer insights into human disease.”
- Line 99: table s1: the authors reported that only one horse had OA. What about the others?
Authors’ response: Thank you for this comment. We apologize that this was not clear. In Table S1, the column on the right provides the reason for the euthanasia, when known. Only one of these animals was so severely lame that euthanasia was elected because of severe pain due to osteoarthritis, on humane grounds. Tables S2 and S3 provide additional information on any degenerative changes observed on the articular surfaces of the joints from which the studied menisci were sourced.
Authors’ action: The following has been added to the Figure legend to provide additional clarity.
“The reason for euthanasia, when known, is indicated. All the collected joints were assessed for articular cartilage and meniscal changes as indicated in Tables S2 and S3”
- Line 100: full-stop should be deleted.
Authors’ response and action: Thank you. It has been corrected.
- Line 101: why did the authors store in saline-soaked gauze and place the tissues at -20°C? The use of saline-soaked gauze could disrupt the tissues.
Authors’ response: When biological tissue is stored in a freezer, there's a risk of dehydration/dessication. Saline-soaked gauzes, combined with sealed plastic bags, help maintain the moisture content of the tissue and prevent desiccation. This method is commonly used in medical settings to preserve tissue samples. It is a standard protocol in our orthopedic research laboratory for many years.
Authors’ action: We have added the following to provide additional clarity:
L148-149
“ ..stored in saline-soaked gauze in sealed plastic bags to avoid dessication ..’.
- Section 2.1: Table s2. What kind of scores did the authors use? They should be added with specific references. Did the authors start from OARSI score or Pauli score etc?
Authors’ response: Thank you for this comment. We referred to our previous paper where we reported this in the submitted manuscript.
Authors’ action: We have now expanded the figure legend to provide more details for the readership.
Supplementary File.
“Table S2. Criteria for the tissue bank stifle joint and meniscus macroscopic assessment scoring.
Macroscopic articular cartilage degeneration: The articular surfaces in the stifle joint were examined for cartilage changes, following the application of India ink, prior to inclusion in the bank. These included the distal femur (medial and lateral femoral condyles and trochlear ridges,) and proximal tibia (medial and lateral tibial plateau) and patella (proximal, middle and distal) as described previously (1).The worst lesion of each area was scored employing a score adapted from a prior study (2). Data for the medial and lateral femorotibial compartments was retrieved from the archive for the present study to pair findings to the studied menisci.
Meniscal macroscopic degeneration: The macroscopic changes (fibrillation and tears) in each of the 3 regions (cranial horn; body and caudal horn), of either the tibial and femoral meniscal surfaces were scored employing a score adapted from Pauli et al. (3) following the application of India ink (1). The meniscus macroscopic ROI score was the sum of the scores for the femoral and tibial surfaces.”
- Dubuc, J., Girard, C., Richard, H., De Lasalle, J., and Laverty, S., Equine meniscal degeneration is associated with medial femorotibial osteoarthritis. Equine Vet J, 2017Jan;50(1):133-140.
- Tiraloche G, Girard C, Chouinard L, Sampalis J, Moquin L, Ionescu M, Reiner A, Poole AR, Laverty S. Effect of oral glucosamine on cartilage degradation in a rabbit model of osteoarthritis. Arthritis Rheum. 2005 Apr;52(4):1118-28
- Pauli C, Grogan SP, Patil S, Otsuki S, Hasegawa A, Koziol J, Lotz MK, D'Lima DD. Macroscopic and histopathologic analysis of human knee menisci in aging and osteoarthritis. Osteoarthritis Cartilage. 2011 Sep;19(9):1132-41.
- Line 117: did the authors use frozen tissues?
Authors’ response: Thank you for asking. We had omitted to indicate that the frozen tissues were thawed.
Authors’ action: We inserted:
L 166“The menisci were thawed in water.”
- Line 119: full-stop is missing.
Authors’ response and action: Thank you for this comment. Suggestions have been followed.
- Figure 1: the sample called B has two sides that match the samples Cr and Ca. It is not clear to me how it was cut.
Authors’ response: We apologize for lack of clarity.
Authors’ action:
We have added:
L179 “ in the centre of” to the figure legend
- Section 2.5: supplier of all antibodies should be added. Dilutions used of the antibodies should be added. I noticed that the antibodies were a gift. This point should be better explained. Are these home-made or commercial antibodies? If home-made, did the authors verify the specificity?
Authors’ response: Some of the details requested were supplied in an Appendix in the original version as the manuscript was long. We have now expanded this information and inserted it in a supplementary methods file to address the reviewers’ concerns, to the best of our ability. The antibodies have been tested in many species in the past. If preferred, we can insert it into the main text.
Authors’ action:
Appendix A
“Immunohistochemistry protocol for C1,2C, NITEGE and DIPEN staining
The NITEGE and DIPEN antibodies were a gift from Dr John Mort RIP. The C1,2C was a gift from Dr Robin Poole. These antibodies were raised in rabbits by the Joint Disease Laboratory, Shriners Hospital for Children (Poole and Mort). McGill University, Montreal, Canada
Slides were deparaffinised in xylene and sequential descending alcohols and then washed in demineralized water (5 mins) and PBS (5 mins). Antigen retrieval was performed by incubating slides with 1% hyaluronidase (type 1-S from bovine testes; Sigma-Aldrich, Saint Louis, USA) in PBS at 37°C for 30 minutes and subsequent washing in PBS (5 mins). The endogenous peroxidase activity was quenched with 3% hydrogen peroxide in methanol (5 mins) at room temperature and the slides were rinsed in PBS (x 2 for 2.5mins). Non-specific secondary antibody marking was blocked with 10% goat serum (#053110, Multicell Wisent, St-Bruno, QC) in PBS, 1% w/v BSA (30 mins). The sections were then incubated overnight at 4°C with NITEGE antibody or 60 minutes at room temperature with DIPEN antibody, diluted 1:250 in PBS and 1% w/v BSA.
When using C1,2C antibody, slides were incubated overnight at 4°C, diluted 1:300 in PBS and 1% w/v BSA. All slides were washed in PBS (x 3 for 2.5 mins) prior to subsequent incubation with a biotinylated goat anti-rabbit secondary antibody (#31822, Thermo Fisher Scientific, Rockford, IL), diluted 1:200 in PBS, 1% w/v BSA (45 mins) at room temperature. The sections were then washed in PBS (x 3 for 2.5 mins) and further incubated with the avidin-biotin complex labeled with peroxidase (Vector Laboratories, California, USA) for 45 minutes. All slides were washed in PBS (x 3 for 2.5 mins) prior to development of the peroxidase reaction, an 8 minutes incubation of the samples with DAB (Vector labolatories, California, USA). All sections were counterstained with a standard hematoxylin protocol and mounted with VectaMount mounting medium (Vector Laboratories, Burlington, ON, Canada).”
- Software used for statistical analysis should be added.
Authors’ response and action:
Line 247-248 “SAS v. 9.4 (Cary, N. C.) software” was added
- Section 3.2: this part should be better explained. The authors discussed of 42 ROI blocks but there are 34 menisci and each meniscus should be divided into three paraffin blocks, for a total of 102 blocks and not 42.
Authors’ response: Thank you for comment. In section 2.6 we presented the challenges encountered when sectioning the mature meniscal tissue. We have now attempted to make this clearer
Authors’ action: The following has been added to the methods:
L 257-259. “From the 34 selected menisci, 104 ROI blocks were sectioned. Subsequently, only ROI blocks that provided enough sections, with minimal sectioning artefacts (see section 2.6), that allowed for comparisons between parameters were included in the final analysis”.
- Line 217, 245, 255: why 38 or 40 or 42? Is this related to table 1?
Authors’ response: We recognize and apologize that the challenges encountered obtaining suitable quality sections without artefacts following the quality control step (section 2.6) made the results section more complex. We remain convinced however that our findings in tissues with spontaneous, non-experimental disease provides important insight.
Authors’ action: We have now attempted to spell out further the limitations. We are open to any additional suggestions.
L 289-290
“A total of 38 meniscal ROI sections were included in the histological analysis. Four sections (39-42Table 1) were excluded because of suboptimal quality for scoring purposes.”
L 345-346
“Generalized NITEGE staining (score 3) was identified in 14% (6/42) of ROIs, (Figure 4 and Table 1).”
L 355-358
“Colocalization assessment was not possible in a limited number of ROIs as occasional poor-quality sections were eliminated from the analysis (Table 1). Colocalization of NITEGE and DIPEN staining was observed in the majority of specimens where it was assessed (64%; 25/39) (Table 1).”
Table 1 figure legend has been expanded:
L 265-269
“Table 1. The meniscal ROIs included in the final analysis, following the quality control step (section 3.2) and the corresponding scores for each of the parameters assessed in site-matched sections. Following the quality control step the listed ROIs were judged to have ample quality sections, with minimal artefacts to permit comparisons across modalities indicates that scoring was not possible in a section. The ROI was included as information was available for other parameters.”
- Table 1 is not clear. It is not clear which horse each ROI corresponds to.
Authors’ response: Thank you for comment.
Authors’ action: We have now inserted a column Horse # into Table 1 that provides the information on the meniscal source (from table S1) into table 1.
- Line 255: um should be corrected.
Authors’ response: We are sorry but we do not understand this request as the lines do not correspond to our submitted version of manuscript. We will be happy to follow up on this matter.
- Lines 328-329: what are these controls?
Authors’ response and action: This was a qualitative assessment as indicated in section 2.6 to compare structure at both ends of the spectrum, health and disease. We have changed the word control to healthy.
- Section 3.9: is there a correlation with age of the horses? Could the authors check?
Authors’ response: The data was not available for 4 of 17 horses and ages spanned from 9-27 (all adults) so we elected not to do this as we did not believe it would be robust data.
Authors’ action: If the reviewer, believes this is very important we can do so.
- It is not clear to me why the authors did not divide and compare the samples based on the presence/absence of OA and tears.
Authors’ response: Thank you for raising this pertinent point. These relationships were explored in an earlier version of the manuscript.
To understand and assess the relationship between meniscal structural ECM changes and molecular degenerative events the meniscal ROI histological cores were arbitrarily divided into 2 groups: control meniscal sections with minimal histological structural changes (score3) and a meniscal degenerative group (histologic score 3). This score was derived from the assumption that meniscal histologic fibrillation (score 1) at the 3 sites assessed would provide a composite score of 3 and would be compatible with normal wear and tear or mild meniscal structural degeneration. Comparisons of extracellular matrix parameters were made between both groups to detect any differences.
The ECM scores were also compared between meniscal ROI with and without macroscopic tears. Finally, the menisci were also categorized into arbitrary groups based on their joint of origin compartment macroscopic OA scores to evaluate the association between meniscal degradation and OA.
No significant associations were identified. As the manuscript was long , the methods complex and the specimens limited, because of the challenges encountered sectioning the specimens we elected not to include all this data in the current version as we thought it already a difficult read. Should this reviewer wish us to incorporate all this data again, we will be happy to do so.
Authors’ action: None
- Lines 359-361: this sentence should be tone down as it is unclear how many horses had OA and how many had tears etc.
Authors’ response: We have removed this statement.
- I am not able to find appendix A, B and C. Authors should check.
Authors’ response: Thank you. Please see response to # 19.

Reviewer 2 Report
Comments and Suggestions for Authors
The authors studied the effects of ECM degradation in meniscal tissues of horses.
Comments
1. The Title should be modified. It should indicate the source of meniscal tissue.
2. Abstract: The authors should indicate the source of meniscal tissue here also.
3. Abstract: Conclusion is rather contradictory. It should be rewritten.
4. Abstract: It is not clear whether 14 specimens (42 ROI) or 17 samples from adult horses (Line 97) or 34 (line 106) were included in the study? This should be clarified.
5. Lines 21-22, 26-28, 37-39, 55-59; 71-74; 77; 87-92; 98-99: These sentences are not clear. They should be rephrased.
6. The authors should avoid unclear word combinations such as “10-year-old specimen”, “loss of ECM NITEGE”,” “MMP cleavage recognized by DIPEN” etc.
7. Introduction: The structure and composition of the meniscus in health and disease should be described in the Introduction section. The term “meniscal intrasubstance” should be defined. The authors should present a scheme of the meniscal structure.
8. Figures 4 and 5: Structure and staining of Controls of healthy tissues and their descriptions should be included.
9. Section 3.9: The authors should present all the Correlation coefficients and p-values in a separate Table.
10. Lines 361-362: The authors should indicate what kind of “significant association” they describe.
11. Conclusion is rather contradictory. It should be rewritten.
Comments on the Quality of English Language1. Lines 21-22, 26-28, 37-39, 55-59; 71-74; 77; 87-92; 98-99: These sentences are not clear. They should be rephrased.
2. The authors should avoid unclear word combinations such as “10-year-old specimen”, “loss of ECM NITEGE”,” “MMP cleavage recognized by DIPEN” etc.
Author Response
Thank you to the Editor and reviewers for time taken to review the manuscript and for the valuable input that has helped improve it.
We apologize as it seems to us that information included in appendices was not perhaps shared with the reviewers on the initial review and may have contributed to some confusion. Supplementary information and appendices are again provided.
Ms. Ref. No.: ijms-2971320
Title: Degradation of Proteoglycans and Collagen in Meniscal Tissue
|
|
The authors studied the effects of ECM degradation in meniscal tissues of horses.
Comments
- The Title should be modified. It should indicate the source of meniscal tissue.
Authors’ response and action: Thank you for suggestion. Changes made.
- Abstract: The authors should indicate the source of meniscal tissue here also.
Authors’ response: Thank you.
Authors’ action: It was mentioned at L 15 in original version that this was equine menisci.
- Abstract: Conclusion is rather contradictory. It should be rewritten.
Authors’ response: Thank you.
Authors’ action: We have modified the conclusion as follows:
Line 35-39
“Proteoglycan and collagen degradation commonly occur superficially in menisci and less frequently centrally. The identification of central meniscal proteoglycan and collagen degradation provides novel insight into central meniscal degeneration. However, further research is needed to elucidate the etiology and sequence of degradative events.”
- Abstract: It is not clear whether 14 specimens (42 ROI) or 17 samples from adult horses (Line 97) or 34 (line 106) were included in the study? This should be clarified.
Authors’ response: Thank you.
Authors’ action: The abstract and manuscript have been modified for additional clarity. Please also see response # 20 and 21 to reviewer 1.
- Lines 21-22, 26-28, 37-39, 55-59; 71-74; 77; 87-92; 98-99: These sentences are not clear. They should be rephrased.
Authors’ response:
Authors’ action: The following modifications were made:
L25-28: “The proteoglycan scores exhibited significant associations with both histologic evaluation (p=0.03) and DIPEN scores (p=0.02). Additionally, a robust positive correlation (p=0.007) was observed between the two aggrecanolysis indicators, NITEGE and DIPEN scores.”
L 35-39: “Proteoglycan and collagen degradation commonly occur superficially in menisci and less frequently centrally. The identification of central meniscal proteoglycan and collagen degradation provides novel insight into central meniscal degeneration. However, further research is needed to elucidate the etiology and sequence of degradative events.”
L 55-60: “An MRI study has revealed that the prevalence of meniscal damage, including meniscal tear, maceration or destruction, in the general population (mean age 62), is 35%[3]. Meniscal tears were observed in 31% of healthy people, with most (77%) considered degenerative horizontal and complex (40% and 37% respectively), tears[3,.4] with the remaining principally traumatic.”
L81-86. “Although biochemical assessments have been conducted to evaluate meniscal collagen content[13, 42], and gene expression analyses have explored collagen synthesis[13-16], the examination of meniscal collagen degradation via the detection of collagenase-generated cleavage products in situ within naturally occurring disease settings has not, to our knowledge, been undertaken in any species.”
L101-108. “Briefly, proteolysis of the aggrecan core protein occurs at the interglobular domain, yielding specific cleavage sites with neoepitopes that can be detected immunohistochemically using antibodies DIPEN and NITEGE, which are produced by MMPs and ADAMTS, respectively (as reviewed by Roughley and Mort[22]). An experimental investigation into cytokine-induced meniscal degradation in sheep revealed greater levels of ADAMTS-mediated aggrecan cleavage (NITEGE) in the inner meniscus, while MMP-driven aggrecanolysis (DIPEN) predominated in the outer meniscus, as evidenced by western blot analysis [15].”
L111-115 “Our previous investigations into naturally occurring meniscal disease in horses have unveiled parallels with human pathology: meniscal tears and lesions are distributed throughout all meniscal regions[25, 26], with the medial meniscus exhibiting the highest frequency of involvement, and the prevalence of disease increases with age[26].”
L127-132: “The aim of our study encompassed two primary objectives: 1) to examine the degradation of equine meniscal extracellular matrix (ECM)proteoglycan and collagen, employing specific antibodies targeting their cleavage sites, and to correlate these findings with site-matched histological analyses from control and naturally occurring disease meniscal tissues; and 2) to characterize the ECM collagen structure within normal meniscal tissue and compare it with sites exhibiting ECM degradation”
L 144-146.:“Information regarding signalment and the origin of the banked stifle joints is provided in Supplementary Table S1.”
- The authors should avoid unclear word combinations such as “10-year-old specimen”, “loss of ECM NITEGE”,” “MMP cleavage recognized by DIPEN” etc.
Authors’ response: Thank you. We have modified the phrases to the best of our abilities.
L384
“A specimen aged ten years”
L 475
“reduction in extracellular matrix NITEGE”
L 467
“MMP-mediated cleavage identified by DIPEN immunostaining”
- Introduction: The structure and composition of the meniscus in health and disease should be described in the Introduction section. The term “meniscal intrasubstance” should be defined. The authors should present a scheme of the meniscal structure.
Authors’ response : Thank you for your suggestions about describing the structure and composition of the meniscus in health and disease. We have now expanded on meniscal structural changes in disease and osteoarthritis in the introduction and referred to an excellent review article on the subject (Englund et al. 2012) and this we hope addresses, in part, some of your suggestions. To expand further we believe is beyond the scope of the current manuscript and would be more fitting for a review.
We have also now defined meniscal intrasubstance degeneration as requested :
Authors’action:
The following has been inserted:
Line 45-53 “OA is a degenerative disease of the joint organ characterized by progressive articular cartilage fibrillation and erosion, the formation of periarticular and central subchondral osteophytes and sclerosis accompanied by inflammation, fibrosis, and pain. The role of the meniscus in OA pathology has been reviewed by Englund et al[2] . Factors such as knee misalignment, obesity, and excessive strain from occupational activities or injury can lead to meniscal damage and tears impairing its function and are risk factors for OA[2]. Articular cartilage loss occurs at areas of meniscal damage, indicating a strong cause-and-effect relationship between the two.”
L 64-67 “
“MRI meniscal intrasubstance abnormality, within the meniscal core, are linear signals confined within the meniscus thought to represent areas of meniscal degeneration or intrasubstance tears.[2]”
Ahmad, R. Intra-substance meniscal changes and their clinical significance: a meta-analysis. Sci Rep 11, 3642 (2021). https://doi.org/10.1038/s41598-021-83181-5
- Figures 4 and 5: Structure and staining of Controls of healthy tissues and their descriptions should be included.
Authors ‘response action: We have changed the order of figures 4 and 5 (now 3 and 4) as an error was made inserting them and modified their legends. We have also included a new figure as requested. (Figure 5).
- Section 3.9: The authors should present all the Correlation coefficients and p-values in a separate Table.
Authors’ response: Since we used the Cochran-Mantel-Haenszel (CMH) test to examine the association between our ordinal scores, it is not possible to provide a correlation value for each test. The test statistic with CMH is a chi-square value and this cannot be converted to a correlation in the usual sense.
Authors’ action: We have removed mention of the word correlation and now use the word association instead.
- Lines 361-362: The authors should indicate what kind of “significant association” they describe.
Authors’ response and action: We have decribed the association as being positive or negative throughout the revised manuscript.
- Conclusion is rather contradictory. It should be rewritten.
Authors’ response and action: See response reviewer 2, # 3.

Round 2
Reviewer 1 Report
Comments and Suggestions for Authors
I have still some questions
Line 26: “positive correlation” is still present.
Lines 261, 311: “um” should be corrected. The Greek letter mu should be used for microns.
Reviewer 2 Report
Comments and Suggestions for Authors
I have no more comments. Accept as is.